# A Comparative Study of Green Inhibitors for Galvanized Steel in Aqueous Solutions

**Vesna Alar, Ivan Stojanović and Dražen Mezdić *** 

Faculty of Mechanical Engineering and Naval Architecture, University of Zagreb, Ivana Lucica 5, 10020 Zagreb, Croatia; vesna.alar@fsb.hr (V.A.); ivan.stojanovic@fsb.hr (I.S.)
* Correspondence: drazen.mezdic@fsb.hr; Tel.: +385-1-6168-360

**Abstract:** This study investigates the morphology, composition and corrosion resistance of hot dip galvanized (HDG) steel bolts in fresh water, 1% NaCl saltwater in the presence of protective compounds L-Tryptophan and three conventional corrosion inhibitors: Inhibitor 1-VCI (amine-carboxylates-based), Inhibitor 2 (based on carboxylate acid), and Inhibitor 3 (water-based, carboxylate acid). Quantitative tests performed include gravimetric analysis and electrochemical techniques, such as linear polarization, Tafel extrapolation, and Electrochemical Impedance Spectroscopy (EIS). Electrochemical measurements determined the polarization resistance $R_p$, corrosion rate $v_{corr}$, and corrosion potential $E_{corr}$. Furthermore, Scanning Electron Microscopy (SEM) and ATR-FTIR spectrometry were used to evaluate and characterize the formed layers on the surface of hot-dip galvanized steel samples. The results reveal that upon corrosion new compounds are formed onto the exposed areas of the treated bolts such as zinc-carbonates, zinc-hydroxides, etc. The presence of these compounds on the surface of the HDG steel bolts hinders the corrosion process by reducing the rate of the anodic and cathodic reactions. The gravimetric test showed that L-tryptophan in distilled water has mass increase, resulting from the formation of white deposits on the sample surface. In fresh water and distilled water, the best corrosion resistance was achieved with Inhibitor 1-VCI, while L-tryptophan showed best results in 1% NaCl solution.

**Keywords:** inhibitors; hot-dip galvanized steel; aqueous solutions

## 1. Introduction

Zinc is one of the most widely used metallic materials in many applications, like coating for steel or as an alloying element in brass and other alloys. Hot-dip galvanizing prevents corrosion by offering barrier and galvanic protection to the base steel. The galvanized coating is metallurgically bonded to the underlying steel, forming an impervious barrier between the steel substrate and the corrosive environment [1]. The use of zinc as a corrosion-resistant coating is usually limited to atmospheric corrosion, while in more aggressive environments it is necessary to protect steel with alternative corrosion protection methods. The corrosion rate of zinc in water and aqueous solutions can be significantly reduced by use of an appropriate corrosion inhibitor [2].

Inorganic anions provide passivation protection to metal surfaces through their incorporation into the oxide layer; the most widely used of these are: chromate ($CrO_4^{2-}$), nitrate ($NO_3^-$), molybdate ($MoO_3^-$), phosphate ($H_2PO_4^-$) and silicates. Recently, the influence of organic, environmentally friendly corrosion inhibitors for zinc protection is increasingly being investigated [3,4]. One of these environmentally friendly corrosion inhibitors is L-tryptophan, an amino acid from the indole group, which has shown good properties in zinc protection in sodium chloride solution at a concentration of 1 $\times 10^{-2}$ mol/L [5,6]. L-tryptophan could be found in protein-rich foods such as beef, chicken, fish and dairy products. This amino acid can also be found in the seeds of the plant Griffonia Simplicifolia,

from which a food additive is made. Its molecular structure consists of an indole ring, nitrogen and oxygen atoms, which are crucial to creating a thin monomolecular film on the metal surface and prevent metal contact with aqueous solutions [7–10]. A thorough examination of the available literature revealed that almost all tested amino acids compounds have been demonstrated to exhibit good ability for use as eco-friendly inhibitors against metal corrosion in different media. However, there is no a universal amino acid compound that is applicable to most metal/solution systems. Hence, the inhibition capability of these compounds is dependent on their molecular structure, their concentration, corrosive medium, metallic surface nature, and on other factors. While, some amino acids can act as effectiveness corrosion inhibitors, some can also show an opposite effect, which accelerates the corrosion process; this all depends on the operating conditions.

The objective of this paper is to study the corrosion behaviour of four green inhibitors for improvement of the corrosion resistance and white rust prevention of hot dipped galvanized steel bolts in fresh water, 1% NaCl saltwater, and distilled water. The water is usually used immediately after the hot-dip galvanizing process for cooling the samples. Distilled water was used as the reference solution without chloride. However, in practice, fresh water or processed water containing about 75 ppm of chloride ions is typically used. As cooling in practice is carried out in bathtubs, there is an increase in chloride ions in water during production, which significantly affects the quality of the zinc coating. Tests were also conducted in 1% NaCl as the most extreme case to see the efficiency of the inhibitor and in such conditions. The bolts are galvanized in perforated metal basket. Once the metallurgical reaction is complete and the fasteners are completely coated, the zinc is drained from the basket. The basket is then spun to remove excess zinc. The bolts are then cooled to room temperature by immersion in water. If the bolts are not properly dried before packing, the zinc corrosion (i.e., white rust) occurs.

## 2. Experimental

The experimental study included gravimetric analysis and visual assessment, electrochemical polarization measurements, electrochemical impedance spectroscopy, SEM and EDX analysis and ATR-FTIR spectrometry. Electrochemical polarization measurements were used to determine the corrosion potential $E_{corr}$, polarization resistance $R_p$, and corrosion rate $v_{corr}$. Electrochemical Impedance Spectroscopy was conducted to determine the resistance of the inhibitor layer. Scanning Electron Microscopy (SEM) and Energy-dispersive X-ray spectroscopy (EDX), as well as Attenuated total reflection Fourier Transform Infrared spectrometry (ATR-FTIR), were used to evaluate and characterize the formed layers on the surface of hot-dip galvanized steel samples.

### 2.1. Materials and Sample Preparation

Zinc coating was achieved by hot immersion at a low temperature. The zinc coating consisted of four phases (Gamma, Delta, Zeta and Eta). Eta ($\eta$) phase is consisting practically of the pure zinc and is formed by simple solidification of the zinc melt (content of the iron is about 0.03 wt.%). Zinc crystallizes in the hexagonal system and is characterized as having relatively high toughness under common temperatures and low hardness. Intermetallic phase zeta ($\zeta$) can be stoichiometrically defined as $FeZn_{13}$. Content of the iron in this phase is about 5–6.2 wt.%. It crystallizes in the coating in base centred monoclinic system. The content of iron in delta phase is 7–11.5 wt.% and intermetallic compounds form the hexagonal crystal structure. Gamma ($\Gamma$) phase can be stoichiometrically expressed as $Fe_5Zn_{21}$ or $FeZn_4$; however most recently as $Fe_{11}Zn_{40}$. It crystallizes in the face-centred cubic system (FCC) with an iron content of 17–19.5 wt.% [11].

The study was conducted on galvanized steel bolts in fresh water, 1% NaCl saltwater, and distilled water, with and without inhibitor. The evaluated inhibitors were L-tryptophan, Inhibitor 1-VCI, Inhibitor 2, and Inhibitor 3. All inhibitors were applied in the testing solution at a concentration of 200 ppm. Before corrosion testing, the measured thickness of the Zn-coating was 40–50 µm using magnetic induction gauge.

Inhibitor 1-VCI (sample 1.1, 1.2, 1.3) is a water-based (amine-carboxylate) corrosion preventive liquid that meets tough antipollution requirements designed as a complete replacement for oil-based preventives for indoor protection of equipment and components. It is not intended for application in chloride containing solutions.

Inhibitor 2 (sample 2.1, 2.2, 2.3) is a chemical cleaner containing corrosion inhibitor based on carboxylate acid, used to remove rust and residue from steel and galvanized steel, as well as from the most existing metals and alloys. Contact with the metal surface creates a passivated layer, which is resistant to corrosion.

Inhibitor 3 (sample 3.1, 3.2, 3.3) is a water-based carboxylate acid inhibitor designed as a complete substitute for corrosion protection with solvent-based coatings. It is applied on the metal surface by spraying or immersion. The protective coating is transparent.

L-tryptophan (sample 4.1, 4.2, 4.3) is an amino acid whose molecular structure consists of an indole ring, which is of particular importance in the field of research of environmentally acceptable corrosion inhibitors. In solutions containing nitrogen and oxygen, it creates a strong adsorption process on the metal surface and protects the metal against corrosion [7].

## 2.2. Gravimetric Testing and Visual Assessment

Gravimetric tests were performed by measuring the mass of the samples before ($m_{before}$) and after 24 h ($m_{after}$) immersion in fresh water, 1% NaCl saltwater, and distilled water, with and without inhibitor [12]. The inhibitors were added to the test solutions at a concentration of 200 ppm. Before the immersion, the samples were degreased in ethanol and dried. Based on measured data the corrosion rate ($v_{corr}$) and inhibition efficiency ($\eta$) was calculated.

The average corrosion rate ($v_{corr}$) was calculated using the following Equation (1):

$$v_{corr} = \frac{\Delta m \times K}{\rho \times A \times t} \tag{1}$$

where $\Delta m$ is the weight loss, $K$ defines the units of the corrosion rate in millimetres per year, $\rho$ is the density of zinc, $A$ is exposed area and $t$ is exposure time [13].

The percentage of inhibition efficiency ($\eta$) was calculated using the Equation (2):

$$\eta(\%) = \frac{\Delta m_0 - \Delta m_i}{\Delta m_0} \times 100 \tag{2}$$

where $\Delta m_0$ and $\Delta m_i$ are the weight loss values in absence and presence of inhibitor [14].

## 2.3. DC Polarization Measurements

The polarization measurements were carried out with a Potentiostat/Galvanostat Model 273 EG & E device (AMATEK, Advanced Measurement Technology, Inc. Berwyn, PA, USA) in a standard three electrode test cell in relation to the reference saturated calomel electrode (SCE), with known interaction potential of +0.242 V versus a standard hydrogen electrode (SHE).

Polarization curves were registered after 600 s of exposure to 400 mL of testing solution in order to allow corrosion potential ($E_{corr}$) stabilization. Polarization resistance ($R_p$) was determined in the potential range of ±20 mV around the corrosion potential using a scan rate of 0.166 mV/s. During the potentiodynamic measurements, the working electrode was polarized to the potential of ±250 mV relative to the corrosion potential and the current ($I_{corr}$) response was measured. Based on measured data unprotected and protected galvanized steel samples, the corrosion rate ($v_{corr}$) and inhibition efficiency ($\eta$) was calculated.

The values of the corrosion rate ($v_{corr}$) were obtained using Equation (3):

$$v_{corr} = \frac{K \times I_{corr} \times EW}{\rho \times A} \tag{3}$$

where $I_{corr}$ is the corrosion current density, $EW$ is the equivalent weight of metal, $\rho$ is the density of zinc, $A$ is the sample area and $K$ is the constant that defines the units of the corrosion rate in millimetres per year [15].

Inhibition efficiency ($\eta$) was calculated using Equation (4):

$$\eta(\%) = \frac{I^0_{corr} - I^i_{corr}}{I^0_{corr}} \times 100 \qquad (4)$$

where $I^0_{corr}$ and $I^i_{corr}$ are the values of corrosion current density in the absence and presence of inhibitor [14].

### 2.4. Electrochemical Impedance Spectroscopy

The protective properties of the inhibitor layer were investigated by electrochemical impedance spectroscopy [16], with a VersaSTAT 3 Potentiostat/Galvanostat EG & G PAR 378 (AMATEK, Advanced Measurement Technology, Inc. Berwyn, PA, USA). The measurements were carried out after 24 h of immersion in the testing solution at room temperature ($23 \pm 2$) °C, with and without inhibitor. The impedance spectra were performed at open circuit potential (OCP) with a 10 mV sinusoidal amplitude. The frequency range was from 100 kHz to 10 mHz. A three-electrode cell including a galvanized steel sample as the working electrode, a saturated calomel electrode (SCE) as the reference electrode and graphite sticks as the auxiliary electrodes were used in the experiments. The surface of the working electrode was 1 cm$^2$. The Solartron Z-View 2.2 software (AMATEK, Advanced Measurement Technology, Inc. Berwyn, PA, USA) was used to interpret data. The corrosion inhibition efficiency ($\eta$) was determined using Equation (5):

$$\eta(\%) = \frac{R^i_{sum} - R^0_{sum}}{R^i_{sum}} \times 100 \qquad (5)$$

where $R^i_{sum}$ and $R^0_{sum}$ are the total resistance in presence and absence of inhibitor [14].

### 2.5. SEM and EDX Analysis and ATR-FTIR Spectrometry

Scanning Electron Microscopy (SEM, OXFORD instruments, Abingdon, UK) and ATR-FTIR spectrometry (Thermo Fiher Scientific, Waltham, MA, USA) were used to evaluate and characterize the formed layers on the surface of hot-dip galvanized steel samples. The surface coverage of the inhibitor layer was observed by a Tescan scanning electron microscope (SEM) equipped with Oxford Instruments energy dispersive spectroscopy (EDX). ATR-FTIR spectrometry was performed at PerkinElmer Spectrum One FTIR Spectrometer [17].

## 3. Results and Discussion

### 3.1. Visual Assessment and Gravimetric Analysis

From the visual assessment results obtained after 24 h immersion of the samples in aqueous solutions with and without inhibitors, it is apparent that Inhibitor 1 (samples 1.1 and 1.2) in water and distilled water showed the best results, that is, no visible corrosion occurrence, whereas in 1% NaCl solution the surface changes were more pronounced, in the form of black spots. On samples protected by inhibitors 2 and 3, the damage was found in all testing solutions, while L-tryptophan resulted in white deposits on the surface of the samples. The results are shown in Table 1.

**Table 1.** Samples of galvanized bolts after 24 h in solutions with and without inhibitors.

| Solution | No. Inhibitor | Inhibitor 1-VCI | Inhibitor 2 | Inhibitor 3 | L-Tryptophan |
|---|---|---|---|---|---|
| Fresh water (samples 1) |  |  |  |  |  |
| Distilled water (samples 2) |  |  |  |  |  |
| 1% NaCl (samples 3) |  |  |  |  |  |

Gravimetric analysis showed that almost all samples lost the weight after 24 h of immersion. The highest mass loss (0.2873 g) occurred on a sample immersed in a 1% NaCl solution treated with Inhibitor 1-VCI (Sample 1.3), which corresponds to visual observation after immersion, possibly due to insufficient concentration of the inhibitor. On the sample treated with L-tryptophan in distilled water, mass growth was observed, which was a result of white deposits on the surface of the screws.

Table 2 shows the gravimetric test results before ($m_{before}$) and after ($m_{after}$) treatment in aqueous solutions and calculated values of inhibition efficiency ($\eta$) and corrosion rate ($v_{corr}$).

**Table 2.** Gravimetric test results before and after treatment in aqueous solutions.

| Sample | $m_{before}$ (g) | $m_{after}$ (g) | $\Delta m$ (g) | $v_{corr}$ (mm/year) | $\eta$ (%) |
|---|---|---|---|---|---|
| 1 | 21.7963 | 21.7909 | 0.0054 | 0.023 | - |
| 2 | 21.8112 | 21.7592 | 0.052 | 0.219 | - |
| 3 | 21.7553 | 21.5931 | 0.1622 | 0.682 | - |
| 1.1 | 21.8969 | 21.8956 | 0.0013 | 0.006 | 75.93 |
| 1.2 | 21.8322 | 21.7711 | 0.0611 | 0.257 | - |
| 1.3 | 21.8685 | 21.5812 | 0.2873 | 1.207 | - |
| 2.1 | 21.8953 | 21.8498 | 0.0455 | 0.191 | - |
| 2.2 | 21.7701 | 21.766 | 0.0041 | 0.017 | 92.12 |
| 2.3 | 21.7867 | 21.7794 | 0.0073 | 0.030 | 95.50 |
| 3.1 | 21.5936 | 21.5867 | 0.0069 | 0.029 | - |
| 3.2 | 21.6011 | 21.5833 | 0.0178 | 0.074 | 65.77 |
| 3.3 | 21.7335 | 21.7312 | 0.0023 | 0.009 | 98.58 |
| 4.1 | 21.8281 | 21.8262 | 0.0019 | 0.008 | 64.81 |
| 4.2 | 21.7714 | 21.7636 | 0.0078 | 0.032 | 85.00 |
| 4.3 | 21.8133 | 21.8119 | 0.0014 | 0.005 | 99.14 |

### 3.2. Open Circuit Potential Measurement (OCP)

Open circuit potential or corrosion potential ($E_{corr}$) was determined in relation to the reference saturated calomel electrode (SCE) on a surface of 1 cm$^2$ in 400 mL of fresh water, distilled water and a 1% NaCl solution. The concentration of each inhibitor was 200 ppm, while 0.816 g of L-tryptophan

was used. The results of corrosion potential measurements are given in Figures 1–3. The most negative values of corrosion potential were measured in 1% NaCl solution, with and without corrosion inhibitors. The results are presented in detail in Tables 3–5.

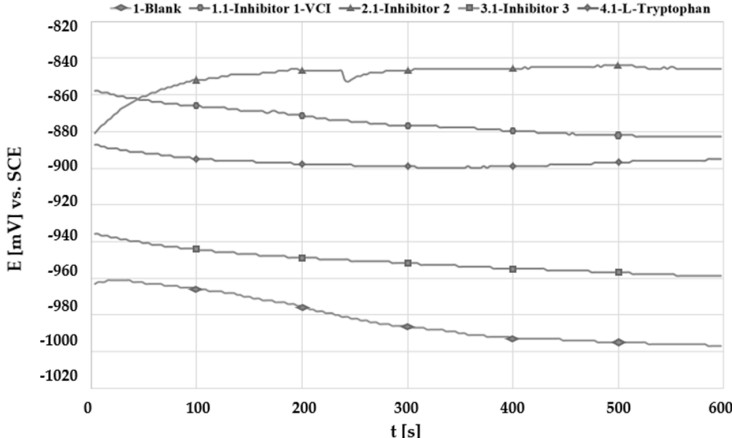

**Figure 1.** Corrosion potential diagram with and without the addition of inhibitors in fresh water.

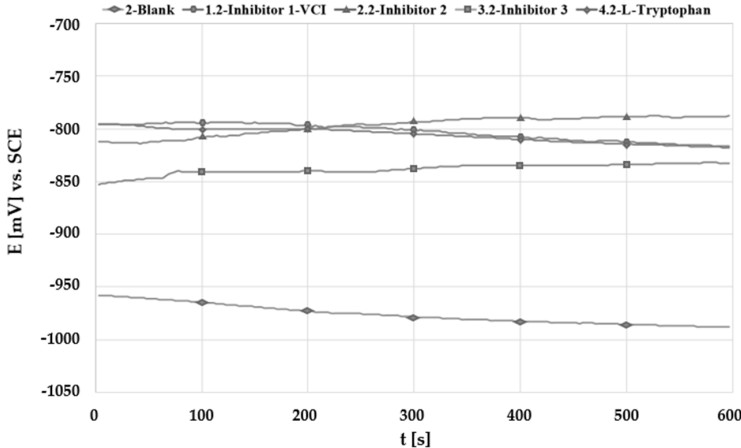

**Figure 2.** Corrosion potential diagram with and without the addition of inhibitors in distilled water.

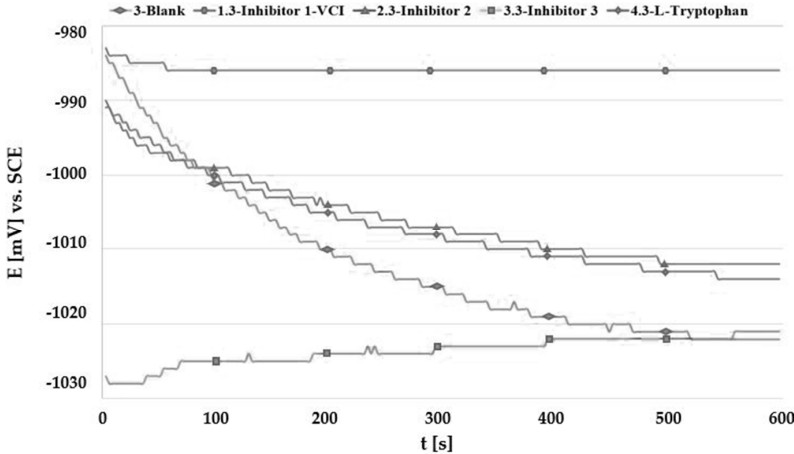

**Figure 3.** Corrosion potential diagram with and without the addition of inhibitors in 1% NaCl.

**Table 3.** Polarization test results in fresh water, at a temperature (23 ± 2) °C, with and without inhibitor.

| Sample | $E_{corr}$ vs. *SCE* (mV) | $\beta_a$ (mV/dec) | $\beta_c$ (mV/dec) | $j_{corr}$ (μA/cm²) | $\eta$ (%) |
|---|---|---|---|---|---|
| 1 | −890 | 97.6 | 288.1 | 21.14 | - |
| 1.1 | −886 | 121.8 | 240.7 | 3.483 | 83.52 |
| 2.1 | −963 | 145.3 | 156.4 | 5.788 | 72.62 |
| 3.1 | −854 | 57.14 | 167.8 | 10.51 | 50.28 |
| 4.1 | −986 | 106.8 | 252.2 | 22.86 | - |

**Table 4.** Polarization test results in distilled water, at a temperature (23 ± 2) °C, with and without inhibitor.

| Sample | $E_{corr}$ vs. *SCE* (mV) | $\beta_a$ (mV/dec) | $\beta_c$ (mV/dec) | $j_{corr}$ (μA/cm²) | $\eta$ (%) |
|---|---|---|---|---|---|
| 2 | −822 | 241.2 | 282.7 | 3.364 | - |
| 1.2 | −811 | 125.6 | 216.4 | 2.166 | 35.61 |
| 2.2 | −839 | 228.5 | 288.1 | 17.13 | - |
| 3.2 | −791 | 57.19 | 268.3 | 6.969 | - |
| 4.2 | −992 | 257.2 | 282.8 | 11.32 | - |

**Table 5.** Polarization test results in 1% NaCl, at a temperature (23 ± 2) °C, with and without inhibitor.

| Sample | $E_{corr}$ vs. *SCE* (mV) | $\beta_a$ (mV/dec) | $\beta_c$ (mV/dec) | $j_{corr}$ (μA/cm²) | $\eta$ (%) |
|---|---|---|---|---|---|
| 3 | −1010 | 36.16 | 197.1 | 17.64 | - |
| 1.3 | −989 | 11.19 | 260.6 | 31.88 | - |
| 2.3 | −1029 | 68.7 | 236.3 | 13.97 | 20.8 |
| 3.3 | −1016 | 36.08 | 187.4 | 16.11 | 8.67 |
| 4.3 | −1023 | 214.1 | 276.4 | 11.83 | 32.94 |

All tested inhibitors showed a slightly positive shift in corrosion potential compared to blank samples in fresh water. This could be due to the formation of corrosion products and/or oxide film on the Zn surface [18]. The same was observed in distilled water. The drop of corrosion potential with time at the begging of testing in 1% NaCl solution, especially pronounced for blank sample, indicates that the chloride ions attacked the Zn-oxide causing its corrosion. Similar behaviour was observed in [18].

The tested inhibitors (1, 2, 3) have large molecular weights and functional polar bonds (C=O, N-N) by which they adsorb on the surface of the metal and protect it from corrosion [19]. L-tryptophan adsorb on the metal surface through the indole ring with π-electrons and nitrogen/oxygen atom with long pair-electrons in its molecule [5].

### 3.3. Linear Polarization Measurements

The polarization resistance ($R_p$) is determined in the potential interval of ±20 mV vs. $E_{corr}$. Higher polarization resistance means lower corrosion current and lower corrosion rate. The results of the polarization resistance tests showed that all analysed inhibitors provide good corrosion resistance in the fresh water. Similar results were shown by other researchers in the paper [20]. For the samples tested in distilled water, Inhibitor 1-VCI showed the highest polarization resistance ($R_p$ = 61.59 kΩ·cm²), while the worst resistance properties were displayed by Inhibitor 2 and L-tryptophan. On the samples tested in 1% NaCl solution, the highest polarization resistance showed L-tryptophan ($R_p$ = 1.013 kΩ·cm²), and the worst properties were achieved by Inhibitor 1-VCI ($R_p$ = 0.216 kΩ·cm²). Linear polarization results are shown in Table 6.

**Table 6.** Polarization test results of corrosion rate and polarization resistance with and without inhibitor in aqueous solutions.

| | Fresh Water | | | Distilled Water | | | 1% NaCl | |
|---|---|---|---|---|---|---|---|---|
| Sample | $v_{corr}$ (mm/year) | $R_p$ (k$\Omega$·cm$^2$) | Sample | $v_{corr}$ (mm/year) | $R_p$ (k$\Omega$·cm$^2$) | Sample | $v_{corr}$ (mm/year) | $R_p$ (k$\Omega$·cm$^2$) |
| 1 | 0.32 | 4.762 | 2 | 0.048 | 26.63 | 3 | 0.263 | 0.466 |
| 1.1 | 0.052 | 11.39 | 1.2 | 0.033 | 61.59 | 1.3 | 0.496 | 0.216 |
| 2.1 | 0.087 | 9.31 | 2.2 | 0.253 | 9.463 | 2.3 | 0.202 | 0.972 |
| 3.1 | 0.159 | 17.94 | 3.2 | 0.104 | 29.66 | 3.3 | 0.248 | 0.416 |
| 4.1 | 0.346 | 8.661 | 4.2 | 0.168 | 19.41 | 4.3 | 0.172 | 1.013 |

### 3.4. Tafel Extrapolation

Tafel extrapolation measurements provide important information about the kinetics of anodic and cathodic reactions [21].

The potentiodynamic polarization curves (Tafel diagrams) of uninhibited and inhibited galvanized steel samples after 1 h in fresh water, distilled water and 1% NaCl solution are given in Figures 4–6.

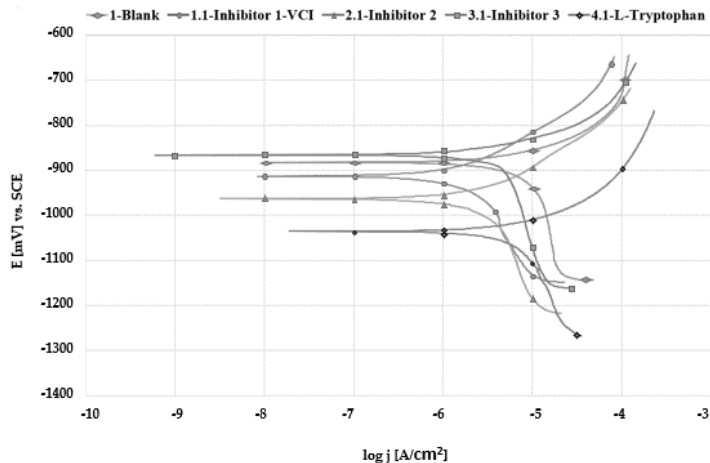

**Figure 4.** Polarization curves of tested inhibitors, compared to unprotected galvanized steel in fresh water.

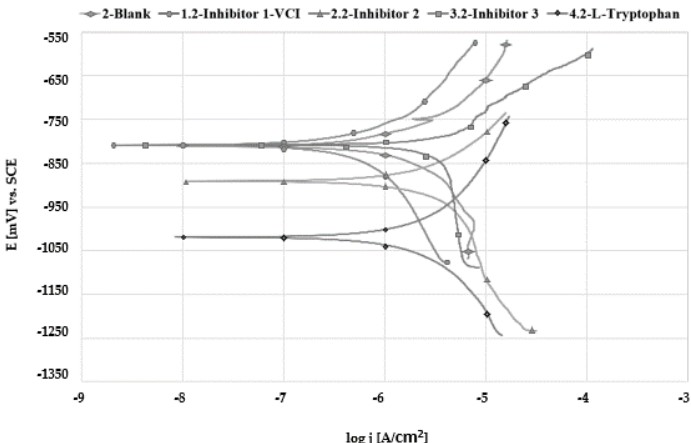

**Figure 5.** Polarization curves of tested inhibitors, compared to unprotected galvanized steel in distilled water.

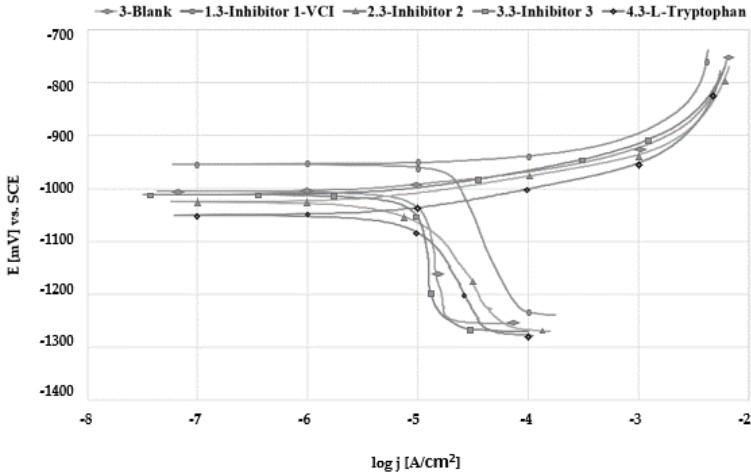

**Figure 6.** Polarization curves of tested inhibitors, compared to unprotected galvanized steel in 1% NaCl solution.

The corrosion parameters, namely corrosion potential ($E_{corr}$), corrosion current density ($j_{corr}$), inhibition efficiency ($\eta$), and anodic and cathodic Tafel slopes ($\beta_a$, $\beta_c$) are listed in Tables 3–5.

The calculated corrosion parameters, corrosion rate ($v_{corr}$), and polarization resistance ($R_p$) with and without inhibitor in aqueous solutions are listed in Table 6.

In fresh water, Inhibitor 3 acted as an anodic inhibitor by reducing anodic reaction and supporting the passivation of Zn surface, while all other inhibitors showed cathodic inhibition behaviour producing insoluble compounds (carbonates and hydroxides) on cathodic sites and preventing hydrogen reduction. In distilled water, Inhibitor 1-VCI and Inhibitor 3 acted as mixed-type inhibitors; while using L-tryptophan and Inhibitor 2, potential displacement to more negative values was observed. In 1% NaCl, the corrosion potential shift to more noble values is observed with application of Inhibitor 1-VCI, indicating its anodic behaviour, while all other inhibitors showed cathodic behaviour. Taking into account that Inhibitor 1-VCI showed anodic behaviour in 1% NaCl solution and that the corrosion rate for Inhibitor 1-VCI was the highest (0.496 mm/year), it can be concluded that its concentration was not sufficient for corrosion prevention forming only partial passivation of the Zn surface, thus accelerating corrosion.

The results of the polarization test showed that the Inhibitor 1-VCI has the best inhibition efficiency in fresh water ($\eta$ = 83.52%), while L-tryptophan is the best in 1% NaCl solution ($\eta$ = 32.94%), which corresponds to the results of the gravimetric test.

Comparing the calculated values of corrosion rate, it was found that Inhibitor 1-VCI showed the best protection properties in fresh water and distilled water, while L-tryptophan showed the best protection in 1% NaCl solution.

*3.5. Electrochemical Impedance Spectroscopy (EIS)*

The obtained Nyquist plots for Inhibitor 1 and 2 have two time constants, one representing the film on the surface and another representing the double layer capacitance, Figure 7. Two time constants indicate the penetration and absorption of the water through a coating which negatively affect the corrosion and adhesion properties at the layer-metal interface [22]. L-tryptophan in 1% NaCl has the highest charge transfer resistance indicating an adsorption mechanism on the surface. Corrosion changes from charge transfer control to diffusion control process, Figure 8. Regarding the Bode plots, when the phase angle is close to 90°, the coating tends to be a pure capacitor, which means good physical barrier properties, whereas if the phase angle is close to 0 the coating tends to be a resistor [23]. From the phase angle of the tested inhibitors, the L-tryptophan showed the best protective properties in 1% NaCl, that is, the phase angle is 90°, Figure 9. The electrochemical behaviour at

the coating/electrolyte interface is the key factor in understanding the protective properties of the inhibitor layer.

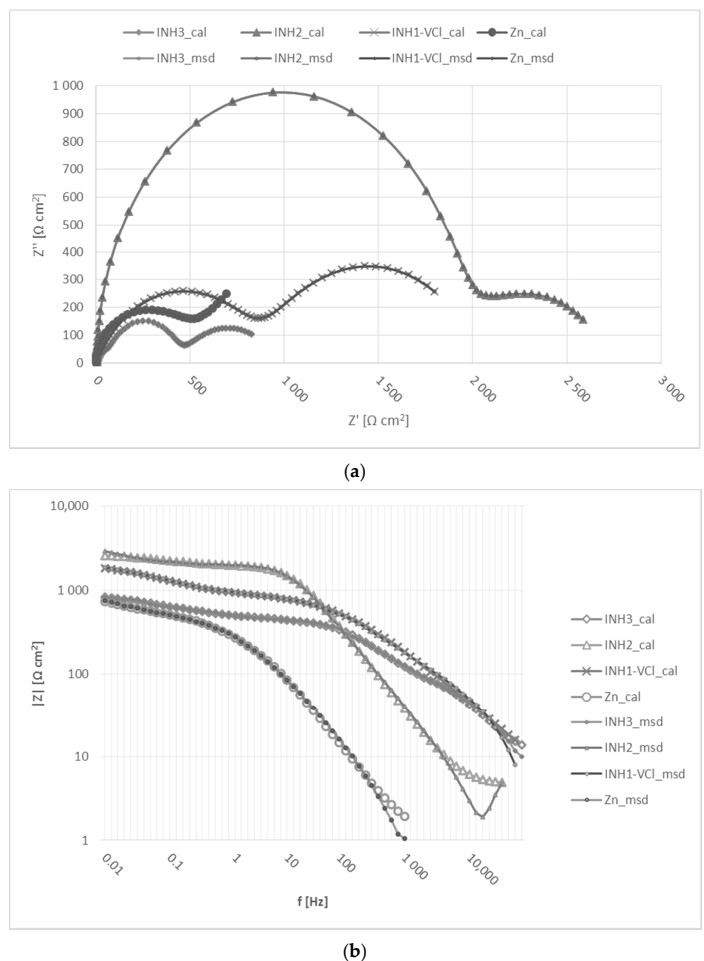

(a)

(b)

**Figure 7.** The Nyquist diagram (**a**) and Bode diagram (**b**) of tested Zn sample, Inhibitor 1-VCI, Inhibitor 2 and Inhibitor 3 after 24 h immersion in 1% NaCl solution.

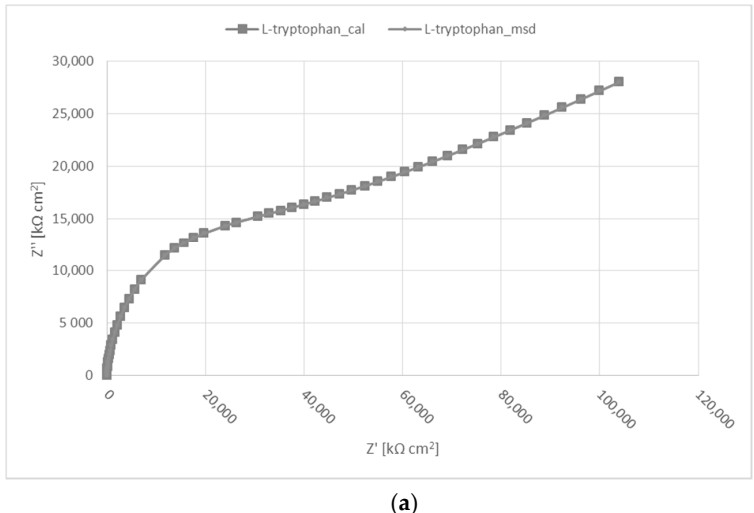

(a)

**Figure 8.** *Cont.*

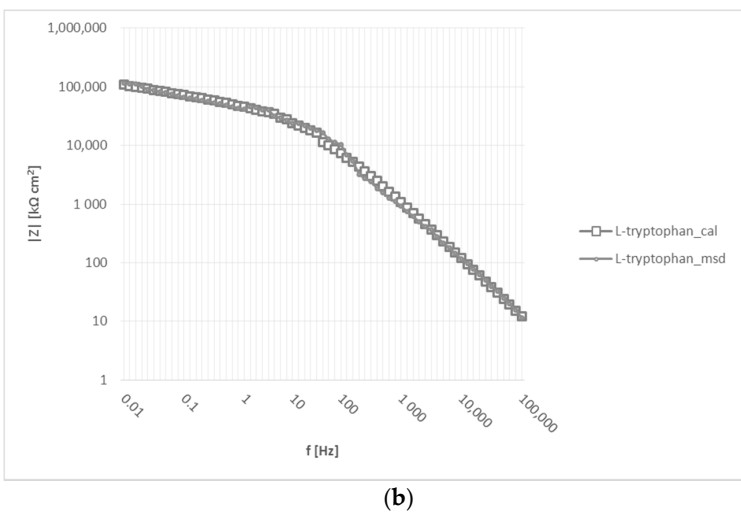

(**b**)

**Figure 8.** The Nyquist diagram (**a**) and Bode diagram (**b**) of tested L-tryptophan after 24 h immersion in 1% NaCl solution.

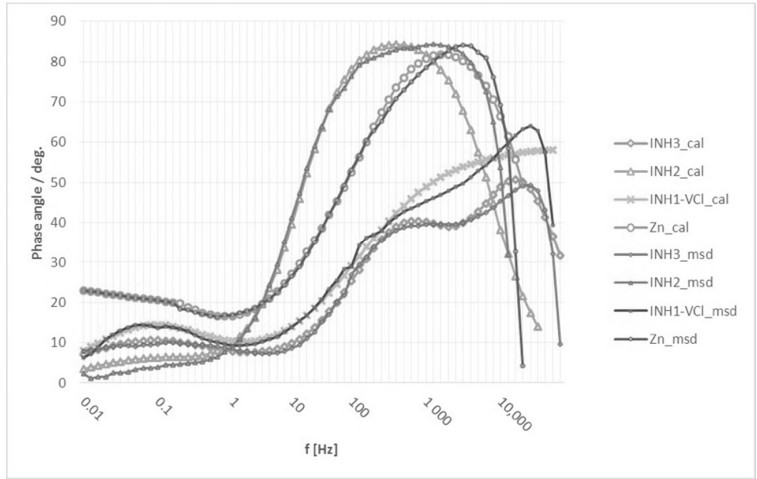

**Figure 9.** The phase angle diagram all of tested samples 24 h immersion in 1% NaCl solution.

The impedance results, measured (msd) and calculated (cal), after 24 h immersion in 1% NaCl solution, expressed by the Nyquist and Bode diagrams are given in Figures 7 and 8. The phase angle diagram is presented in Figure 9.

After 24 h of exposure to testing solutions, the impedance spectra were analysed by fitting with a suitable equivalent circuit model, given in Figure 10. The first model shown in Figure 10a with Warburg element (W) describes the diffusion limited impedance by a layer (film, coating) undergoing corrosion at the metal surface. It is used for fitting the results of L-tryptophan and Zn sample. At high frequencies, the Warburg impedance is small, since diffusing reactants do not have to move very far. At low frequencies, the reactants have to diffuse farther, increasing the Warburg impedance [24]. The second and third circuit model elements shown are as follows: $R_s$ = solution resistance, $R_{coat}$ = coating resistance, $R_{ct}$ = charge transfer resistance, $C_{coat}$ = coating capacitance, and $C_{dl}$ = coating double layer capacitance. The charge transfer resistance and double layer capacitance elements appear for a coating system with corrosion occurring on the metal surface [25,26]. The second equivalent electric circuit shown in Figure 10b is used for Inhibitor 2, describing the porous coating systems, meaning that electrolyte has pass through inhibitor layer and reached the substrate. The third equivalent circuit shown in Figure 10c is used for Inhibitor 1. This equivalent circuit R(CR)(CR) usually indicates that the electrolyte homogeneously diffuses into coatings, causing uniformly distributed reaction sites at

the interface [27], which indicates that relatively rapid corrosion may occur when exposed to chloride environments [25]. The fourth equivalent circuit shown in Figure 10d is used for Inhibitor 3, which exhibits the lowest anti-corrosion performance in 1% NaCl. The three time-constant equivalent circuit models indicate the formation of zinc corrosion product layer (patina) on the surface described by: $R_{cpl}$ = corrosion product layer resistance and $C_{cpl}$ = corrosion product layer capacitance. Also, lower *n* values of Inhibitor 1 and Inhibitor 3 in 1% NaCl indicate a heterogeneous surface and confirm the anodic or cathodic inhibitor behaviour on the zinc surface.

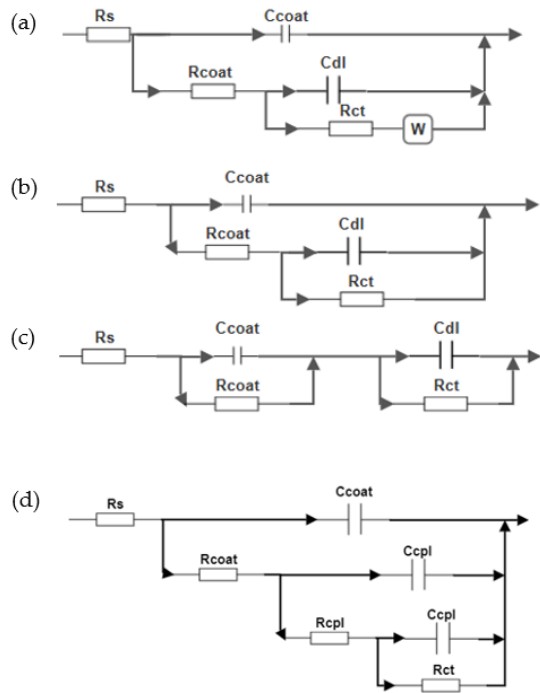

**Figure 10.** Electrical equivalent circuits for: (**a**) L-tryptophan and Zn sample; (**b**) Inhibitor 2; (**c**) Inhibitor 1; and (**d**) Inhibitor 3.

After 24 h, both in fresh water and 1% NaCl, L-tryptophan formed thick and hard removable deposit on the surface of the galvanized steel samples, acting as a barrier and thus resulting in highest layer resistance. These white deposits of L-tryptophan changed the surface of galvanized steel and had a negative influenced on the functionality of fasteners, which is not allowed. The coverage of the thick surface layer of the L-tryptophan product also made obtaining the impedance spectra more difficult [28,29]. For the other three tested inhibitors, Inhibitor 1-VCI showed best protective properties ($R_{sum}$ = 3.11 × 10^5 Ω·cm$^2$) in fresh water, while in 1% NaCl the best corrosion protection was delivered by Inhibitor 2 ($R_{sum}$=2.73 × 10^3 Ω·cm$^2$).

The EIS data, namely solution resistance ($R_s$), inhibitor layer resistance ($R_{coat}$), charge transfer resistance ($R_{ct}$), corrosion product layer resistance ($R_{cpl}$), constant phase element of the inhibitor layer ($CPE_1$) and constant phase element of the double layer ($CPE_2$) with empirical constant $n_1$ and $n_2$, capacitance ($C$), Warburg element ($W$), total resistance ($R_{sum}$) and error of the fitting procedure ($\chi^2$) are given in Table 7.

**Table 7.** EIS results after 24 h immersion in fresh water and 1% NaCl, with and without inhibitor.

| Sample | $R_s$ | $R_{coat}$ | $R_{ct}$ | $R_{cpl}$ | $R_{sum}$ | $C_{coat}$ ($\mu$F/cm$^2$) | CPE$_1$ | | CPE$_2$ | | $W$ ($\Omega$cm$^2$s$^{-1/2}$) | $\chi^2$ | $\eta$ (%) |
|---|---|---|---|---|---|---|---|---|---|---|---|---|---|
| | | | (k$\Omega$·cm$^2$) | | | | $Q_1$ ($\mu$F/cm$^2$) | $n_1$ | $Q_2$ ($\mu$F/cm$^2$) | $n_2$ | | | |
| Zn in fresh water | 0.245 | 1.73 | 0.937 | - | 2.91 | - | 0.916 | 1 | 30 | 0.75 | - | 0.02 | - |
| Zn in 1% NaCl | 0.012 | 0.236 | 0.277 | - | 0.525 | - | 27.1 | 1 | 221 | 0.76 | $3.89 \times 10^{-3}$ | 0.03 | - |
| Inh. 1 in fresh water | 0.721 | 74.4 | 236 | - | 311 | - | 0.311 | 1 | 2.24 | 0.91 | - | 0.03 | 99.06 |
| Inh. 1 in 1% NaCl | 0.029 | 0.746 | 1.19 | - | 1.97 | - | 15.6 | 0.75 | 1480 | 0.61 | - | 0.02 | 73.35 |
| Inh.2 in fresh water | 0.171 | 39.2 | 32.1 | - | 71.47 | - | 0.182 | 1 | 1.36 | 1 | - | 0.18 | 95.93 |
| Inh.2 in 1% NaCl | 0.002 | 1.96 | 0.767 | - | 2.73 | - | 11.7 | 1 | 2780 | 0.72 | - | 0.09 | 80.77 |
| Inh.3 in fresh water | 0.022 | 90.9 | 20.7 | - | 112 | - | 0.53 | 1 | 23.8 | 1 | - | 0.17 | 97.40 |
| Inh.3 in 1% NaCl | 0.011 | 0.072 | 0.517 | 0.367 | 0.967 | 0.855 | 21.58 | 0.8 | 4092 | 0.56 | - | 0.01 | 45.71 |
| L-trypt. in fresh water | 0.011 | 95,100 | 239,000 | - | 334,100 | - | $6.15 \times 10^{-3}$ | 1 | $3.22 \times 10^{-5}$ | 1 | $6.61 \times 10^{-15}$ | 0.09 | 99.99 |
| L-trypt. in 1% NaCl | 0.11 | 8210 | 33,500 | - | 41,710 | - | $0.29 \times 10^{-3}$ | 0.96 | $1.14 \times 10^{-3}$ | 0.93 | $4.81 \times 10^{-8}$ | 0.07 | 99.99 |

EIS results showed that the Inhibitor 1-VCI and L-tryptophan has the best inhibition efficiency in fresh water ($\eta > 99\%$), while L-tryptophan gives the best efficiency in 1% NaCl solution ($\eta = 99.99\%$), which corresponds to gravimetric test and polarization test results.

### 3.6. SEM and EDX Analysis

SEM analysis showed irregularities on all examined samples, i.e., an uneven layer of zinc coating on the surface or chloride depositions from the aqueous solutions. On the samples treated with L-tryptophan, sedimentation of white clusters was visible on the surface. EDX analysis of the tested samples in aqueous solutions with and without inhibitors revealed the chemical composition of adsorbed layers on the galvanized steel samples. L-tryptophan treated samples were found to contain oxygen, which is very important for the process of adsorption of the inhibitor compound on the metal surface. The presence of oxygen in EDX analysis was also confirmed on samples protected with vapour phase corrosion Inhibitor 1-VCI, showing that both corrosion inhibitors act as an adsorption inhibitor. EDX of Inhibitor 3 showed chloride content in surface layer which can cause porosity in the protective film. SEM and EDX view of inhibitor layer after 24 h in fresh and distilled water is given in Figures 11 and 12.

### 3.7. ATR-FTIR Spectrometry

FTIR spectra of tested inhibitors on galvanized steel samples in fresh water are given in Figure 13. The IR spectra of Inhibitor 1-VCI showed several strong and strong to medium peaks which belong to carboxylic acid. The C=O stretch appears at 1504 cm$^{-1}$ and the C-O stretch at 1390 cm$^{-1}$. The 163 cm$^{-1}$ difference between asymmetric and symmetric stretching vibrations corresponds to bidentate binding of carboxylate group to Zn. The observed low intensity vibrations at 471 cm$^{-1}$ correspond to Zn-O stretching vibration.

The IR spectra of Inhibitor 2 showed several strong and strong to medium peaks which belong to carboxylic acid. The C=O stretch appears at 1639 cm$^{-1}$ and the C-O stretch at 1103 cm$^{-1}$. The 536 cm$^{-1}$ difference between asymmetric and symmetric carboxylate stretching vibrations corresponds to unidentate binding of carboxylate group to Zn. The low intensity vibrations at 574 cm$^{-1}$ correspond to Zn-O stretching vibration.

The IR spectra of Inhibitor 3 did not show the exact type of bonding of the inhibitor to Zn. The bonding type in these acids (long-chain fatty acid,) is not ordinary (monodentate or bidentate); thus, it is not easy to describe by IR spectra. Absorption of the inhibitor to Zn could be achieved by several different mechanisms. One of the possibilities is that the bonding is formed in unoccupied spaces of acid chains.

The IR spectra of L-tryptophan showed the strong and broad absorption band in the range of 3600 cm$^{-1}$ to 3000 cm$^{-1}$ which corresponds to asymmetric and symmetric stretching vibration of aqua molecules. Bands in the range of 3392 cm$^{-1}$ and 3161 cm$^{-1}$ belong to the N-H stretches of NH$_2$ group of tryptophan. The weak bands in the 2936–2906 cm$^{-1}$ range are attributed to the CH$_2$ vibrations [28]. In complexes, tryptophan ligands are coordinated to the metal ion as unidentate by carboxylic group. The (COO-)asym peak is located at 1621 cm$^{-1}$ for Zn complexes. The (COO-)sym peak is observed at 1408 for Zn complexes. The low-intensity bands in the region of 600–400 cm$^{-1}$ are attributed to Zn-N and Zn-O vibration.

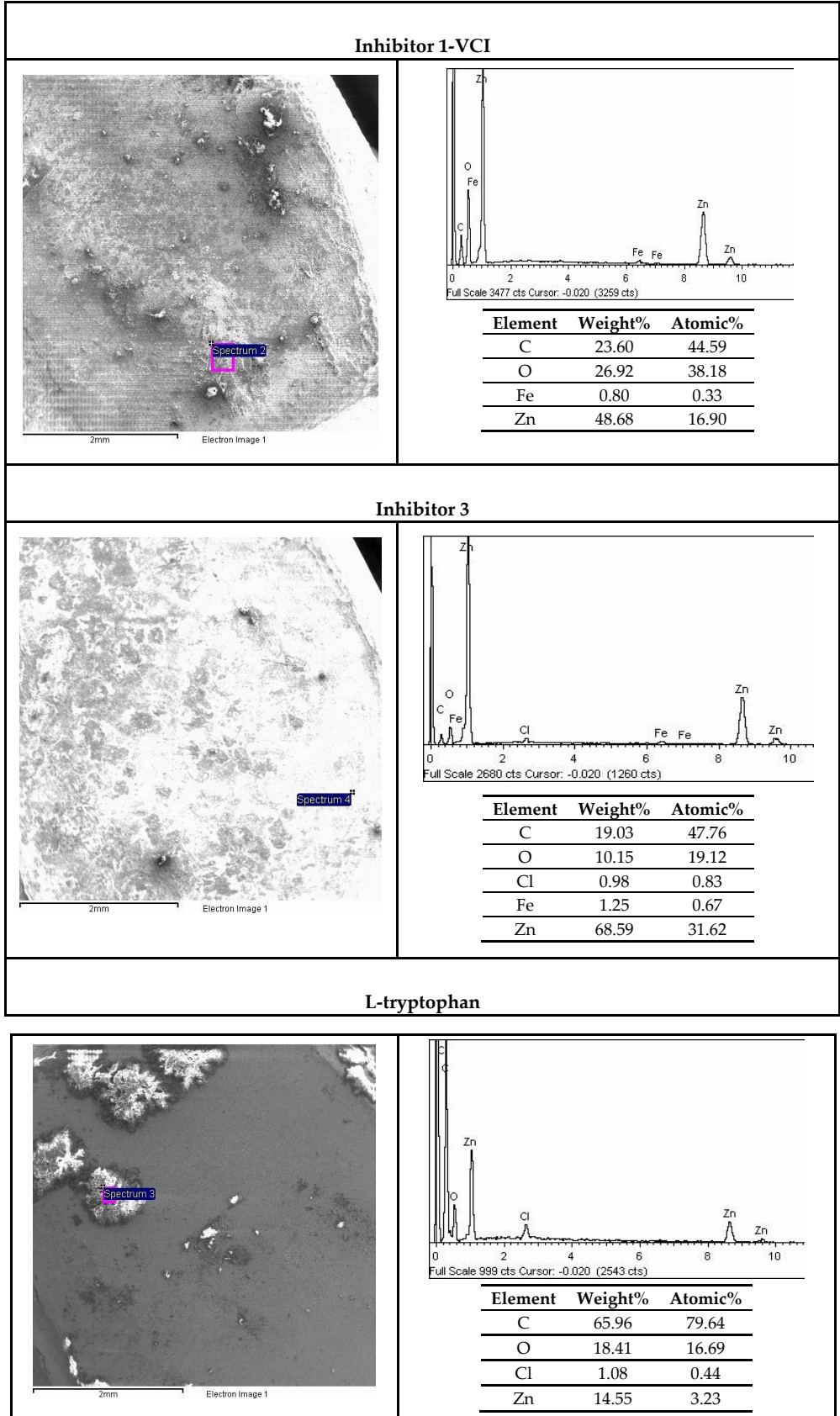

**Figure 11.** SEM and EDX view of inhibitor layer after 24 h in fresh water.

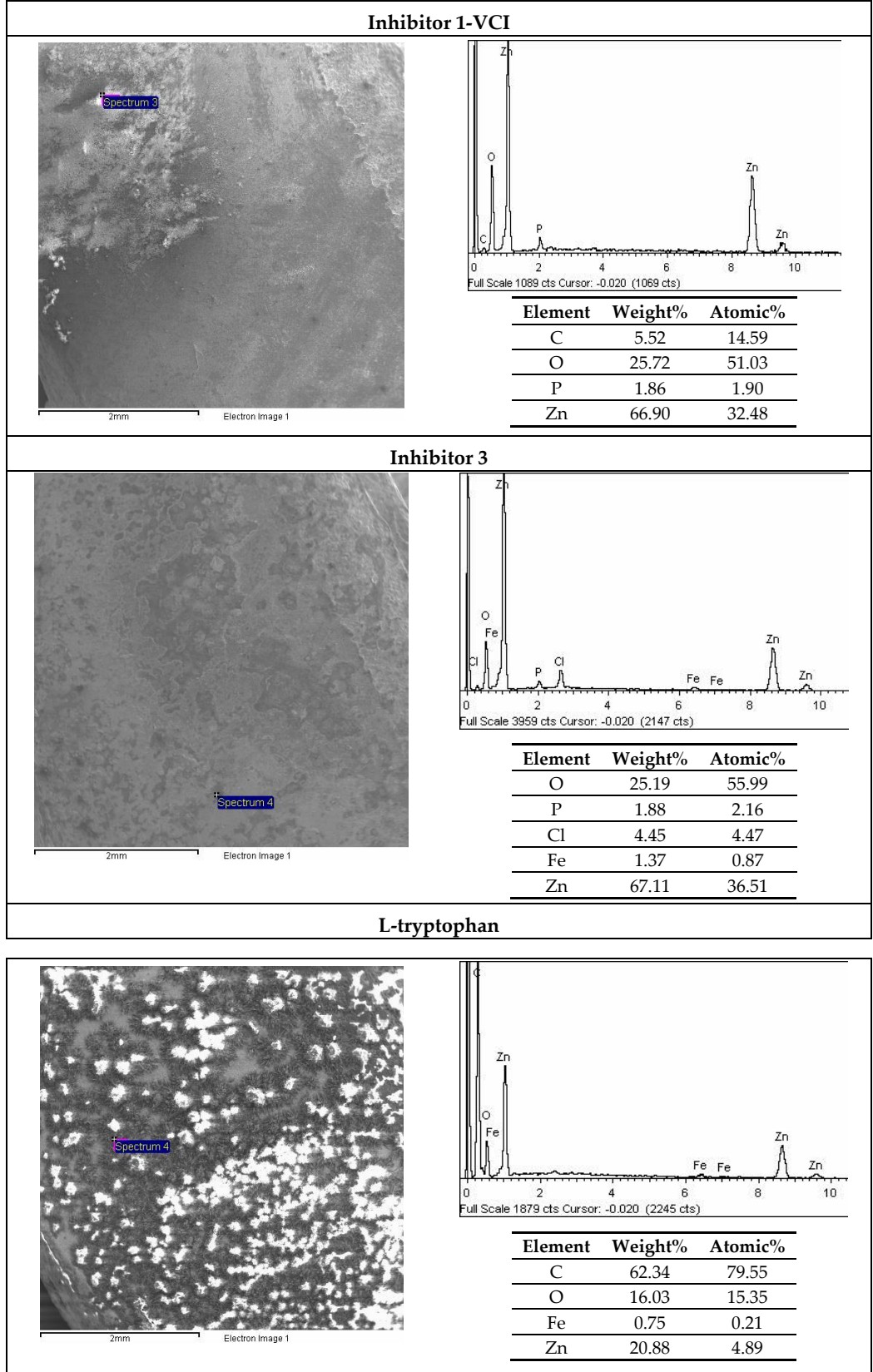

**Figure 12.** SEM and EDX view of inhibitor layer after 24 h in distilled water.

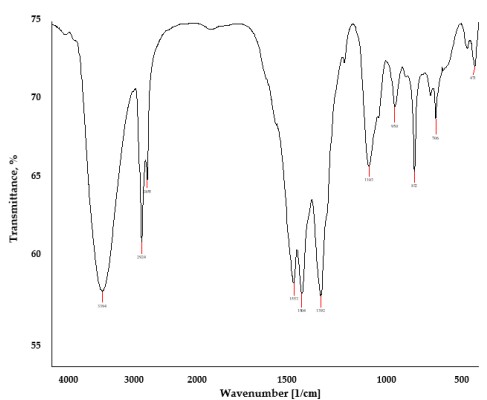

**Inhibitor 1-VCI**

| Maximum cm⁻¹ | Functional Group | |
|---|---|---|
| 471 | Zn-O | stretch |
| 832 | C-H | stretch |
| 1103 | C-O | stretch |
| 1392 | C-O | stretch |
| 1504 | C=O | stretch |
| 2924 | C-H | stretch |
| 3394 | O-H | stretch |

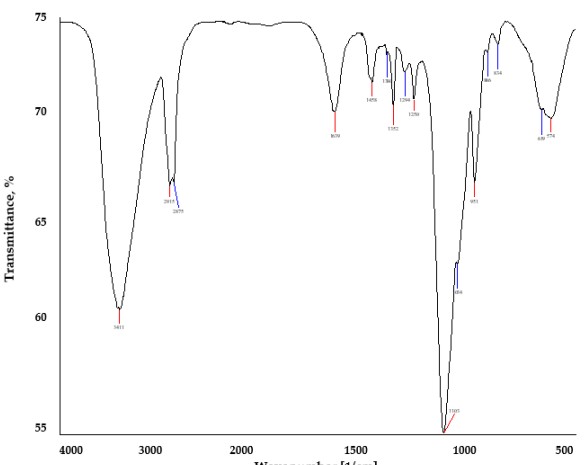

**Inhibitor 2**

| Maximum cm⁻¹ | Functional Group | |
|---|---|---|
| 574 | Zn-O | stretch |
| 619 | C-H | stretch |
| 1034 | C-O | stretch |
| 1103 | C-O | stretch |
| 1639 | C=O | stretch |
| 2915 | C-H | stretch |
| 3411 | O-H | stretch |

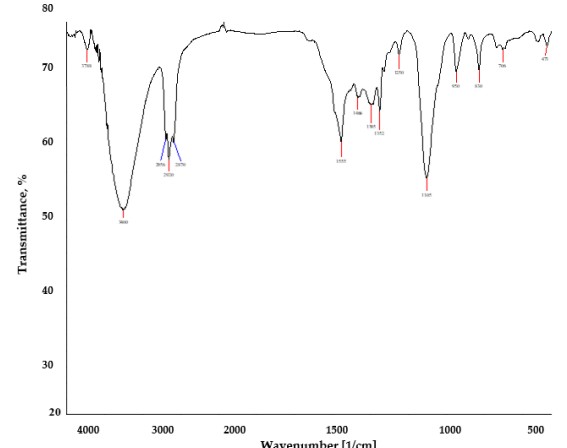

**Inhibitor 3**

| Maximum cm⁻¹ | Functional Group | |
|---|---|---|
| 473 | Zn-O | stretch |
| 830 | C-H | stretch |
| 950 | C-O | stretch |
| 1105 | C-O | stretch |
| 1555 | C=O | stretch |
| 2870 | C-H | stretch |
| 2920 | C-H | stretch |
| 2956 | O-H | stretch |
| 3400 | O-H | stretch |

**Figure 13.** *Cont.*

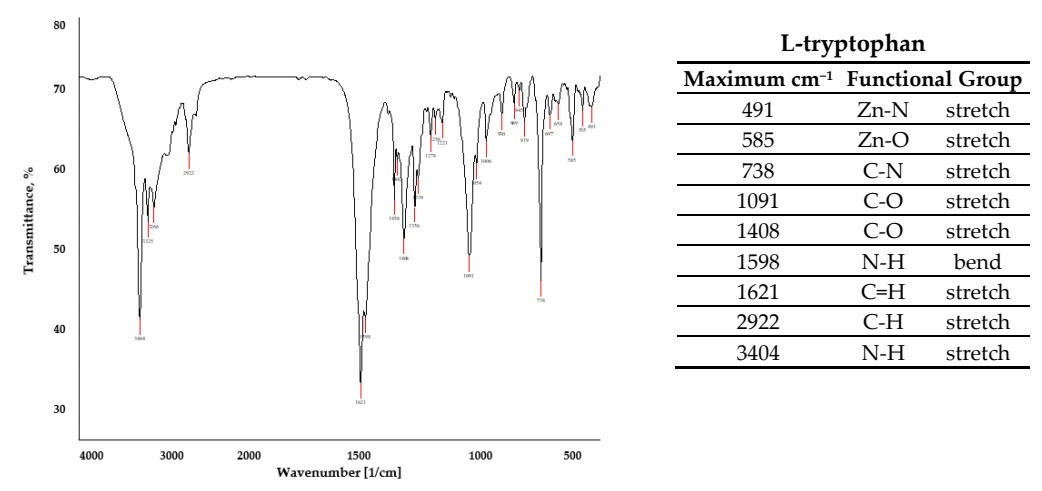

| L-tryptophan | | |
|---|---|---|
| **Maximum cm⁻¹** | **Functional Group** | |
| 491 | Zn-N | stretch |
| 585 | Zn-O | stretch |
| 738 | C-N | stretch |
| 1091 | C-O | stretch |
| 1408 | C-O | stretch |
| 1598 | N-H | bend |
| 1621 | C=H | stretch |
| 2922 | C-H | stretch |
| 3404 | N-H | stretch |

**Figure 13.** FTIR spectra of tested inhibitors on galvanized steel samples in fresh water.

## 4. Conclusions

The inhibition of corrosion is a complex phenomenon and the efficiency of inhibitors depends on a variety of parameters and on the interplay between various competitive effects. The results of this study can be summarized as follows:

(1) Gravimetric testing determined the mass loss of almost all samples after 24 h immersion in aqueous solutions with and without inhibitors. Only the sample treated with L-tryptophan in distilled water showed a mass increase, resulting from the formation of white deposits on the sample surface.

(2) Electrochemical DC tests showed that the best corrosion resistance in fresh water and distilled water was achieved with Inhibitor 1-VCI, while L-tryptophan showed best results in 1% NaCl solution.

(3) Electrochemical impedance spectroscopy showed that L-tryptophan provided the highest layer resistance, but had a negatively influence on the surface functionality of the fasteners and their aesthetic requirements. Inhibitor 1-VCI showed the best protective properties in fresh water and distilled water without changing surface condition of the bolts.

(4) SEM and EDX analysis of galvanized steel samples showed higher oxygen content in L-tryptophan inhibitor film, which is very important for the adsorption process and the formation of a protective inhibitor layer on the zinc coating.

(5) FTIR spectra of the tested inhibitors on galvanized steel samples showed that bonds of inhibitors 1-VCI, Inhibitor 2 and Inhibitor 3 belong to the carboxylic acid group, whereas L-tryptophan belongs to the amino acids.

(6) L-tryptophan acted more as an anodic than a cathodic inhibitor in all tested media.

(7) L-tryptophan can be successfully used to protect zinc coating in solution with increased concentration of chloride ions.

(8) The inhibition efficiency results obtained by three different testing methods, i.e., gravimetric testing, polarization testing and electrochemical impedance spectroscopy, correspond to each other under the same measurement conditions.

**Author Contributions:** Conceptualization, V.A.; methodology, I.S.; software, D.M.; validation, V.A., I.S. and D.M.; formal analysis, V.A.; investigation, I.S. and D.M.; data curation, D.M.; writing—original draft preparation, I.S. and D.M.; writing—review and editing, V.A.; visualization, D.M. All authors have read and agreed to the published version of the manuscript.

**Funding:** This research received no external funding.

**Conflicts of Interest:** The authors declare no conflict of interest.

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
