# Peer review of "A Comparative Study of Green Inhibitors for Galvanized Steel in Aqueous Solutions"

_metals, doi:10.3390/met10040448_

Round 1
Reviewer 1 Report
Comments and Suggestions for Authors
The authors report the investigation of four types of inhibitors for galvanized steel in aqueous solutions. They confirmed the anti-corrosive properties using gravimetric analysis and electrochemical techniques. Finally they evaluated how the anti-corrosive properties depends on a variety of parameters and on the interplay bettween various competentive effects. This work is relatively complete. Minor revision are suggested to publish on this journal.
I only have a one comment:
"The C=O stretch appears at 1103 cm-1?
Author Response
Review 1
Dear reviewer,
Thank you for you kind review and comments that helped to improve our manuscript. Thank you very much for your effort.
In the following pdf, we give a reply to your comments.

Reviewer 2 Report
The manuscript presents a comparative study of four different environmentally-friend inhibitors for zinc in different electrolytes.
The work is well organized, the experimental procedure adequate and the obtained results support the conclusions, although no explanation is put forward to explain why there are changes in the inhibiting mechanism for the different solutions.
To be published the authors should address the following comments:
- Figures have very low quality - it is hard to distinguish between the different curves. The use of colours would help. Moreover, in Figure 3 there are a series of plateaux on the curves, that should be corrected.
- Some explanation should be advanced to why the presence of inhibition 1 in saltwater leads to an increase in the corrosion rate compared to the situation without inhibitor (1.207 vs 0.68 mm/y)
- Which values of ba and bc used for computing Rp? Those obtained from the polarization curves? Is so these experiments should be presented firs or the fact mentioned.
- Finally, the experimental data should be overlayed on the impedance plots (Bode end Nyquist) to give some information on the quality of the fitting (the qui2 is very low).
Author Response

(The authors gave the same response as above.)

Reviewer 3 Report
In general, the manuscript describes an important topic with long-running applications, but several questions arose. What are the reasons for all electrochemical measurements in all investigated solutions? In this case, all charged molecules will have different potentials, and comparing all the results will be problematic. Perhaps it is better to remove from the solution, rinse with deionized water, dry and immerse in deionized water and measure all the electrochemistry? What is the definition of fresh water? What was the chemical composition of fresh water? and what is the difference from deionized water? What was the reason for the study of four different inhibitors? The molecular formulas of these components will help. There is no information about what this white deposit was on bolts after immersion in solutions in the presence of inhibitors? What is the main cause of inhibition: physical sorption on the surface or chelation of salts with an organic compound. What is the mechanism of corrosion? Is it corrosion due to chlorine ions? If yes, why is it corrosion in deionized water? Were these experiments with degassed solutions? or were solubilized oxygen and carbon dioxide present? were they the main cause of corrosion?
How were samples prepared for SEM analysis? They are removed from the test solutions, washed with clean deionized water, dried and stored in a desiccator before use? Sample preparation is very important for this analysis.
Author Response

(The authors gave the same response as above.)

Round 2
Reviewer 3 Report
Authors did answer to all my comments